# LOCAL CRITIC TRAINING OF DEEP NEURAL NETWORKS

## ABSTRACT

This paper proposes a novel approach to train deep neural networks by unlocking the layer-wise dependency of backpropagation training. The approach employs additional modules called local critic networks besides the main network model to be trained, which are used to obtain error gradients without complete feedforward and backward propagation processes. We propose a cascaded learning strategy for these local networks. In addition, the approach is also useful from multi-model perspectives, including structural optimization of neural networks, computationally efficient progressive inference, and ensemble classification for performance improvement. Experimental results show the effectiveness of the proposed approach and suggest guidelines for determining appropriate algorithm parameters.

## 1 INTRODUCTION

In recent days, deep learning has been remarkably advanced and successfully applied in numerous fields (LeCun et al., 2015). A key mechanism behind the success of deep neural networks is that they are capable of extracting useful information progressively through their layered structures. It is an increasing trend that more and more complex deep neural network structures are developed in order to solve challenging real-world problems, e.g., He et al. (2016b). Training of deep neural networks is based on backpropagation in most cases, which basically works in a sequential and synchronous manner. During the feedforward pass, the input data is processed through the hidden layers to produce the network output; during the feedback pass, the error gradient is propagated back through the layers to update each layer's weight parameters. Therefore, training of each layer has dependency on all the other layers, which causes the issue of *locking* (Jaderberg et al., 2017). This is undesirable in some cases, e.g., a system consisting of several interacting models, a model distributed across multiple computing nodes, etc.

There have been attempts to remove the locking constraint. In Carreira-Perpinan & Wang (2014), the method of auxiliary coordinates (MAC) is proposed. It replaces the original loss minimization problem with an equality-constrained optimization problem by introducing an auxiliary variable for each data and each hidden unit. Then, solving the problem is formulated as iteratively solving several sub-problems independently. A similar approach using the alternating direction method of multipliers (ADMM) is proposed in Taylor et al. (2016). It also employs an equality-constrained optimization but with different auxiliary variables, so that resulting sub-problems have closed form solutions. However, these methods are not scalable to deep learning architectures such as convolutional neural networks (CNNs).

The method proposed in Jaderberg et al. (2017), called decoupled neural interface (DNI), directly synthesizes estimated error gradients, called synthetic gradients, using an additional small neural network for training a layer's weight parameters. As long as the synthetic gradients are close to the actual backpropagated gradients, each layer does not need to wait until the error at the output layer is backpropagated through the preceding layers, which allows independent training of each layer. However, this method suffers from performance degradation when compared to regular backpropagation (Czarnecki et al., 2017a). The idea of having additional modules supporting the layers of the main model is also adopted in Czarnecki et al. (2017a), where the additional modules are trained to approximate the main model's outputs instead of error gradients. Due to this, however, the method does not resolve the issue of update locking, and in fact, the work does not intend to design a non-sequential learning algorithm.

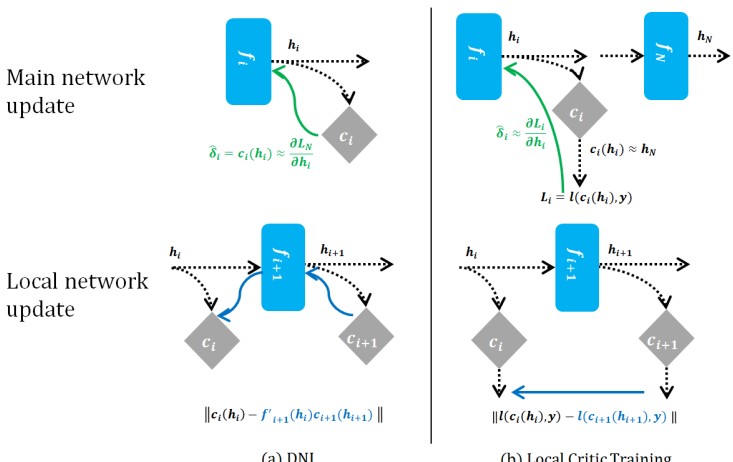

Figure 1: Learning processes of DNI (Jaderberg et al., 2017) and the proposed local critic training. The black, green, and blue arrows indicate feedforward passes, an error gradient flow, and loss comparison, respectively.

In this paper, we propose a novel approach for non-sequential learning, called *local critic training*. The key idea is that additional modules besides the main neural network model are employed, which we call *local critics*, in order to indirectly deliver error gradients to the main model for training without backpropagation. In other words, a local critic located at a certain layer group is trained in such a way that the derivative of its output serves as the error gradient for training of the corresponding layers' weight parameters. Thus, the error gradient does not need to be backpropagated, and the feedforward operations and gradient-descent learning can be performed independently. Through extensive experiments, we examine the influences of the network structure, update frequency, and total number of local critics, which provide not only insight into operation characteristics but also guidelines for performance optimization of the proposed method.

In addition to the capability of implementing training without locking, the proposed approach can be exploited for additional important applications. First, we show that applying the proposed method automatically performs structural optimization of neural networks for a given problem, which has been a challenging issue in the machine learning field. Second, a progressive inference algorithm using the network trained with the proposed method is presented, which can adaptively reduce the computational complexity during the inference process (i.e., test phase) depending on the given data. Third, the network trained by the proposed method naturally enables ensemble inference that can improve the classification performance.

## 2 PROPOSED APPROACH

### 2.1 LOCAL CRITIC TRAINING

The basic idea of the proposed approach is to introduce additional local networks, which we call local critics, besides the main network model, so that they eventually provide estimates of the output of the main network. Each local critic network can serve a group of layers of the main model by being attached to the last layer of the group. The proposed architecture is illustrated in Figure 1, where $f_i$ is the $i$th layer group (containing one or more layers), $h_i$ is the output of $f_i$, and $h_N$ is the final output of the main model having $N$ layer groups:

$$h_i = f_i(h_{i-1}) \tag{1}$$

$c_i$ is the local critic network for $f_i$, which is expected to approximate $h_N$ based on $h_i$, i.e.,

$$c_i(h_i) \approx h_N \tag{2}$$

Then, this can be used to approximate the loss function of the main network, $L_N = l(h_N, y)$, which is used to train $f_i$, by

$$L_i = l(c_i(h_i), y) \tag{3}$$

for $i = 1, ..., N - 1$, i.e.,

$$L_i \approx L_N \tag{4}$$

where $y$ is the training target and $l$ is the loss function such as cross-entropy or mean-squared error. Then, the error gradient for training $f_i$ is obtained by differentiating $L_i$ with respect to $h_i$, i.e.,

$$\delta_i = \frac{\partial L_i}{\partial h_i} \tag{5}$$

which can be used to train the weight parameters of $f_i$, denoted by $\theta_i$, via a gradient-descent rule:

$$\theta_i \leftarrow \theta_i - \eta \, \delta_i \, \frac{\partial h_i}{\partial \theta_i} \tag{6}$$

where $\eta$ is a learning rate. Note that the final layer group $h_N$ does not require a local critic network and can be trained using the regular backproagation because the final output of the main network is directly available. Therefore, the update of $f_i$ does not need to wait until its output $h_i$ propagates till the end of the main network and the error gradient is backpropagated; it can be performed when the operations from (2) to (5) are done. For $c_i$, we usually use a simple model so that the operations through $c_i$ are simpler than those through $f_{i+1}$ till $f_N$.

While the dependency of $f_i$ on $f_j$ ($j > i$) during training is resolved in this way, there still exists the dependency of $c_i$ on $f_j$ ($j > i$), because training $c_i$ requires its ideal target, i.e., $h_N$, which is available from $f_N$ only after the feedforward pass is complete. In order to resolve this problem, we use an indirect, cascaded approach, where $c_i$ is trained so that its training loss targets the training loss for $c_{i+1}$[1]:

$$L_{c_i} = l(L_i, L_{i+1}) \tag{7}$$

In other words, training of $c_i$ can be performed once the loss for $c_{i+1}$ is available.

Figure 1 compares the proposed architecture with the existing DNI approach that also employs local networks besides the main network to resolve the issue of locking (Jaderberg et al., 2017). In DNI, the local network $c_i$ directly estimates the error gradient, i.e.,

$$c_i(h_i) \approx \frac{\partial L_N}{\partial h_i} \tag{8}$$

so that each layer group of the main model can be updated without waiting for the forward and backward propagations in the subsequent layers. And, to update $c_i$, the error gradient for $f_{i+1}$ estimated by $c_{i+1}$ is backpropagated through $f_{i+1}$ and is used as the (estimated) target for $c_i$. Therefore, all the necessary computations in the forward and backward passes can be locally confined. The performance of the two methods will be compared in Section 3.

## 2.2 STRUCTURAL OPTIMIZATION

In many cases, determining an appropriate structure of neural networks for a given problem is not straightforward. This is usually done through trial-and-error, which is extremely time-consuming. There have been studies to automate the structural optimization process (Cortes et al., 2017; Feng & Darrell, 2015; Kwok & Yeung, 1997; Reed, 1993), but this issue still remains very challenging.

In deep learning, the problem of structural optimization is even more critical. Large-sized networks may easily show overfitting. Even if large networks may produce high accuracy, they take significantly large amounts of memory and computation, which is undesirable especially for resource-constrained cases such as embedded and mobile systems. Therefore, it is highly desirable to find an optimal network structure that is sufficiently small while the performance is kept reasonably good.

During local critic training, each local critic network is trained to estimate the output of the main network eventually. Therefore, once the training of the proposed architecture finishes, we obtain different networks that are supposed to have similar input-output mappings but have different structures and possibly different accuracy, i.e., multiple sub-models and one main model (see Figure 2b). Here, a sub-model is composed of the layers on the path from the input to a certain hidden layer

---

[1]We found that this is more effective than directly forcing $c_i$ to approximate $c_{i+1}$ using $L_{c_i} = l(c_i(h_i), c_{i+1}(h_{i+1}))$.

---

**Algorithm 1:** Progressive inference

**Input:** data $x$, threshold $t$
**Model:** sub-model $c_i$, main-model $f$
Initialize: $classification = 0$.
**for** $i = 1$ **to** $N - 1$ **do**
    **if** $max$ softmax$(c_i(x)) > t$ **then**
        $classification = argmax$ softmax$(c_i(x))$
        $break$
    **end if**
**end for**
**if** $classification == 0$ **then**
    # if all sub-models are not confident
    $classification = argmax$ softmax$(f(x))$
**end if**

---

and its local critic network. Among the sub-models, we can choose one as a structure-optimized network by considering the trade-off relationship between the complexity and performance.

It is worth mentioning that our structural optimization approach can be performed instantly after training of the model, whereas many existing methods for structural optimization require iterative search processes, e.g., Zoph & Le (2017).

### 2.3 PROGRESSIVE INFERENCE

We propose another simple but effective way to utilize the sub-models obtained by the proposed approach for computational efficiency, which we call *progressive inference*. Although small sub-models (e.g., sub-model 1) tend to show low accuracy, they would still perform well for some data. For such data, we do not need to perform the full feedforward pass but can take the classification decision by the sub-models. Thus, the basic idea of the progressive inference is to finish inference (i.e., classification) with a small sub-model if its confidence on the classification result is high enough, instead of completing the full feedforward pass with the main model, which can reduce the computational complexity. Here, the softmax outputs for all classes are compared and the maximum probability is used as the confidence level. If it is higher than a threshold, we take the decision by the sub-model; otherwise, the feedforward pass continues. The proposed progressive inference method is summarized in Algorithm 1[2].

### 2.4 ENSEMBLE INFERENCE

In recent deep learning systems, it is popular to use ensemble approaches to improve performance in comparison to single models, where multiple networks are combined for producing final results, e.g., He et al. (2016a); Szegedy et al. (2015). The sub-models and main model obtained by applying the proposed local critic training approach can be used for ensemble inference. Figure 4a depicts how the sub-models and the main model can work together to form an ensemble classifier. We take the simplest way to combine them, i.e., summation of the networks' outputs.

## 3 EXPERIMENTS

We conduct extensive experiments to examine the performance of the proposed method in various aspects. We use the CIFAR-10 and CIFAR-100 datasets (Krizhevsky, 2009) with data augmentation. We employ a VGG-like CNN architecture with batch normalization and ReLU activation functions, which is shown in Figure 2a. Note that this structure is the same to that used in Czarnecki et al. (2017a). It has three local critic networks, thus four layer groups that can be trained independently are formed (i.e., $N=4$). The local critic networks are also CNNs, and their structures are kept as

---

[2]Our method shares some similarity with the anytime prediction scheme (Larsson et al., 2017; Huang et al., 2018) that produces outputs according to the given computational budget. However, ours does not require particular network structures (such as multi-scale dense network (Huang et al., 2018) or FractalNet (Larsson et al., 2017)) but works with generic CNNs.

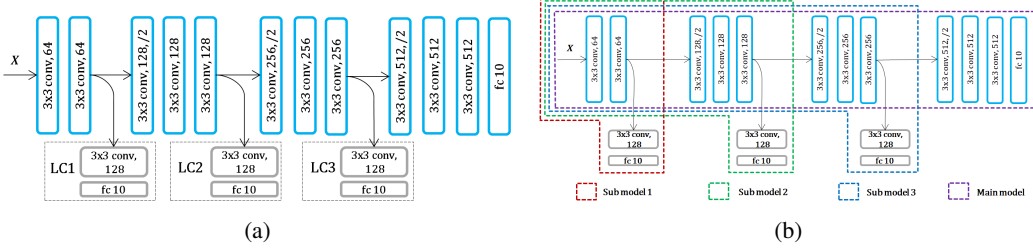

(a)                                                                                      (b)

Figure 2: (a) Network structure of the proposed approach using three local networks for CIFAR-10. LC1, LC2, and LC3 are local critic networks, each of which contains one convolutional layer. For CIFAR-100, the final fc10 layers of the main network and the local critic networks are replaced with fc100. (b) Sub-models and main model obtained by the proposed approach.

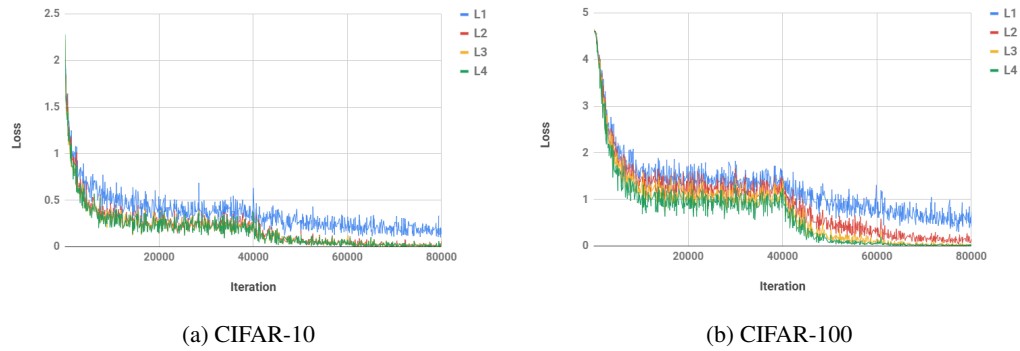

(a) CIFAR-10                                                    (b) CIFAR-100

Figure 3: Training loss values of the main model and each sub-model with respect to the training iteration.

simple as possible in order to minimize the computational complexity for computing the estimated error gradient given by (5).

We use the stochastic gradient descent with a momentum of 0.9 for the main network and the Adam optimization with a fixed learning rate of $10^{-4}$ for the local networks. The L2 regularization is used with $5 \times 10^{-4}$ for the main network. For the loss functions in (3) and (7), the cross-entropy and the L1 loss are used, respectively, which is determined empirically. The batch size is set to 128, and the maximum training iteration is set to 80,000. The learning rate for the main network is initialized to 0.1 and dropped by an order of magnitude after 40,000 and 60,000 iterations. The Xavier method is used for initialization of the network parameters. All experiments are performed using TensorFlow. We conduct all the experiments five times with different random seeds and report the average accuracy.

### 3.1 PERFORMANCE EVALUATION

Figure 3 shows how the loss values of the main network and each local critic network, i.e., $L_i$ in (3), evolve with respect to the training iteration. The graphs show that the local critic networks successfully learn to approximate the main network's loss with high accuracy during the whole training process. The local critic network farthest from the output side (i.e., $L_1$) shows larger loss values than the others, which is due to the inaccuracy accumulated through the cascaded approximation.

The classification performance of the proposed local critic training approach is evaluated in Table 1. For comparison, the performance of the regular backpropagation, DNI (Jaderberg et al., 2017), and critic training (Czarnecki et al., 2017a) is also evaluated. Although the critic training method is not for removing update locking, we include its result because it shares some similarity with our approach, i.e., additional modules to estimate the main network's output. In all three methods, each additional module is composed of a convolutional layer and an output layer. In the case of the proposed method, we test different numbers of local critic networks. Figure 2a shows the structure

Table 1: Average test accuracy (%) of backpropagation (BP), DNI (Jaderberg et al., 2017), critic training (Czarnecki et al., 2017a), and proposed local critic training (LC). The numbers of local networks used are shown in the parentheses. The standard deviation values are also shown.

| Dataset | BP | DNI (3) | Critic (3) | LC (1) | LC (3) | LC (5) |
|---|---|---|---|---|---|---|
| CIFAR-10 | 93.93 $\pm 0.20$ | 64.86 $\pm 0.42$ | 91.92 $\pm 0.30$ | 92.06 $\pm 0.20$ | 92.39 $\pm 0.09$ | 91.38 $\pm 0.20$ |
| CIFAR-100 | 75.14 $\pm 0.18$ | 36.53 $\pm 0.64$ | 69.07 $\pm 0.25$ | 73.61 $\pm 0.31$ | 69.91 $\pm 0.50$ | 63.53 $\pm 0.24$ |

Table 2: Average test accuracy (%) with respect to the number of layers in the local critic networks. $[a, b, c]$ means that the numbers of convolutional layers in LC1, LC2, and LC3 are $a$, $b$, and $c$, respectively.

| Dataset | [1,1,1] (default) | [3,3,3] | [5,5,5] | [3,2,1] | [1,2,3] | [5,4,3] | [3,4,5] |
|---|---|---|---|---|---|---|---|
| CIFAR-10 | 92.39 $\pm 0.09$ | 92.36 $\pm 0.22$ | 91.72 $\pm 0.19$ | 92.07 $\pm 0.21$ | 92.20 $\pm 0.12$ | 92.10 $\pm 0.16$ | 91.90 $\pm 0.16$ |
| CIFAR-100 | 69.91 $\pm 0.50$ | 70.02 $\pm 0.29$ | 70.34 $\pm 0.16$ | 70.06 $\pm 0.64$ | 69.81 $\pm 0.33$ | 70.87 $\pm 0.40$ | 69.93 $\pm 0.56$ |

with three local critic networks. When only one local network is used, it is located at the place of LC2 in Figure 2a. When five local networks are used, they are placed after every two layers of the main network.

When compared to the result of backpropagation, the proposed approach successfully decouples training of the layer groups at a small expense of accuracy decrease (note that the performance of the proposed method can be made closer to that of backpropagation using different structures, as will be shown in Tables 2 and Figure 4b). The degradation of the accuracy and standard deviation of our method is larger for CIFAR-100, which implies that the influence of gradient estimation is larger for more complex problems. When more local critic networks are used, the accuracy tends to decrease more due to higher reliance on predicted gradients rather than true gradients, while more layer groups can be trained independently. Thus, there exists a trade-off between the accuracy and unlocking effect. The DNI method shows poor performance as in (Czarnecki et al., 2017a). The proposed method shows performance improvement by 0.4% and 0.9% over the critic training method, both with three local networks, for the two datasets, respectively, which are found to be statistically significant using Mann-Whitney tests at a significance level of 0.05. This shows the efficacy of the cascaded learning scheme of the local networks in our method.

## 3.2 STRUCTURES OF LOCAL CRITIC NETWORKS

We examine the influence of the structures of the local critic networks in our method. Two aspects are considered, one about the influence of the overall complexity of the local networks and the other about the relative complexities of the local networks for good performance. For this, we change the number of convolutional layers in each local critic network, while keeping the other structural parameters unchanged.

The results for various structure combinations of the three local critic networks are shown in Table 2. As the number of convolutional layers increase for all local networks (the first three cases in the table), the accuracy for CIFAR-100 slightly increases from 69.91% (with one convolutional layer) to 70.02% (three convolutional layers) and 70.34% (five convolutional layers), whereas for CIFAR-10 the accuracy slightly decreases when five convolutional layers are used. A more complex local network can learn better the target input-output relationship, which leads to the performance improvement for CIFAR-100. For CIFAR-10, on the other hand, the network structure with five convolutional layers seems too complex compared to the data to learn, which causes the performance drop.

Next, the numbers of layers of the local networks are adjusted differently in order to investigate which local networks should be more complex for good performance. The results are shown in the last four columns of Table 2. Overall, it is more desirable to use more complex structures for the local networks closer to the input side of the main model. For instance, LC1 and LC3 are supposed to learn the relationship from $h_1$ to $h_4$ and that from $h_3$ to $h_4$, respectively. More layers

Table 3: Average test accuracy (%) with respect to the update frequency of local critic networks.

| Dataset | 1/1 | 1/2 | 1/3 | 1/4 | 1/5 |
|---|---|---|---|---|---|
| CIFAR-10 | 92.39 $\pm 0.09$ | 91.91 $\pm 0.19$ | 91.78 $\pm 0.18$ | 91.57 $\pm 0.12$ | 91.35 $\pm 0.17$ |
| CIFAR-100 | 69.91 $\pm 0.50$ | 67.99 $\pm 0.49$ | 67.76 $\pm 0.19$ | 66.74 $\pm 0.41$ | 66.39 $\pm 0.39$ |

Table 4: Average test accuracy (%) of the sub-models produced by local critic training and the networks trained by regular backpropagation.

| Dataset | BP sub 1 | LC sub 1 | BP sub 2 | LC sub 2 | BP sub 3 | LC sub 3 |
|---|---|---|---|---|---|---|
| CIFAR-10 | 74.46 $\pm 0.91$ | 85.24 $\pm 0.49$ | 88.03 $\pm 0.87$ | 90.53 $\pm 0.15$ | 92.05 $\pm 0.24$ | 92.29 $\pm 0.09$ |
| CIFAR-100 | 47.58 $\pm 1.10$ | 55.39 $\pm 0.57$ | 61.79 $\pm 0.92$ | 63.62 $\pm 0.31$ | 67.81 $\pm 0.22$ | 67.54 $\pm 0.70$ |

are involved from $h_1$ to $h_4$ in the main network, so the mapping that LC1 should learn would be more complicated, requiring a network structure with sufficient modeling capability.

### 3.3 Periodic update of local critic networks

A way to increase the efficiency of the proposed approach is to update the local critic networks not at every iteration but only periodically. This may degrade the accuracy but has two benefits. First, the amount of computation required to update the local networks can be reduced. Second, the burden of the communication between the layer groups also can be reduced. These benefits will be more significant when the local networks have larger sizes.

For the default structure shown in Figure 2a, we compare different update frequency in Table 3. It is noticed that the accuracy only slightly decreases as the frequency decreases. When the update frequency is a half of that for the main network (i.e., 1/2), the accuracy drops by 0.48% and 1.92% for the two datasets, respectively. Then, the decrease of the accuracy is only 0.56% for CIFAR-10 and 1.60% for CIFAR-100 when the update frequency decreases from 1/2 to 1/5.

### 3.4 Structural optimization

Table 4 compares the performance of the sub-models, and Table 5 shows the complexities of the sub-models in terms of the amount of computation for a feedforward pass and the number of weight parameters. A larger network (e.g., sub-model 3) shows better performance than a smaller network (e.g., sub-model 1), which is reasonable due to the difference in learning capability with respect to the model size. The largest sub-model (sub-model 3) shows similar accuracy to the main model (92.29% vs. 92.39% for CIFAR-10 and 67.54% vs. 69.91% for CIFAR-100), while the complexity is significantly reduced. For CIFAR-10, the computational complexity in terms of the number of floating-point operations (FLOPs) and the memory complexity are reduced to only about 30% (15.72 to 4.52 million FLOPs, and 7.87 to 2.26 million parameters), as shown in Table 5. If an absolute accuracy reduction of 1.86% (from 92.39% to 90.53%) is allowed by taking sub-model 2, the reduction of complexity is even more remarkable, up to about one ninth.

In addition, the table also shows the accuracy of the networks that have the same structures with the sub-models but are trained using regular backpropagation. Surprisingly, such networks do not easily reach accuracy comparable to that of the sub-models obtained by local critic training, particularly for smaller networks (e.g., 74.46% vs. 85.24% with sub-model 1 for CIFAR-10). We think that joint training of the sub-models in local critic training helps them to find better solutions than those reached by independent regular backpropagation.

Therefore, these results demonstrate that a structurally optimized network can be obtained at a cost of a small loss in accuracy by local critic training, which may not be attainable by trial-and-error with backpropagation.

Table 5: FLOPs required for a feedforward pass and numbers of model parameters in the sub-models and main model for CIFAR-10. Note that sub-model 2 has less FLOPs and parameters than sub-model 1 due to the pooling operation in sub-model 2.

| model | FLOP | # of parameters |
|---|---|---|
| Sub-model 1 | 2.85M | 1.42M |
| Sub-model 2 | 1.76M | 0.88M |
| Sub-model 3 | 4.52M | 2.26M |
| Main model | 15.72M | 7.87M |

Table 6: Average FLOPs and accuracy of progressive inference for test data of CIFAR-10 when the threshold is set to 0.9 or 0.95.

| | FLOP | Accuracy (%) |
|---|---|---|
| Progressive inference $_{(0.9)}$ | 2.90M | $91.18_{\pm 0.10}$ |
| Progressive inference $_{(0.95)}$ | 3.05M | $91.75_{\pm 0.16}$ |
| Main model | 15.72M | $92.39_{\pm 0.09}$ |

### 3.5 PROGRESSIVE INFERENCE

We apply the progressive inference algorithm shown in Algorithm 1 to the trained default network for CIFAR-10 with the threshold set to 0.9 or 0.95. The results are shown in Table 6. The feedforward pass ends at different sub-models for different test data, and the average FLOPs over all test data are shown. When the threshold is 0.9, with only a slight loss of accuracy (92.39% to 91.18%), the computational complexity is reduced significantly, which is only 18.45% of that of the main model. When the threshold increases to 0.95, the accuracy loss becomes smaller (only 0.64%), while the complexity reduction remains almost the same (19.40% of the main model's complexity).

### 3.6 ENSEMBLE INFERENCE

The results of ensemble inference using the sub-models and main model are shown in Figure 4b. Using an ensemble of the three sub-models, we observe improved classification accuracy (92.68% and 70.86% for the two datasets, respectively) in comparison to the main model. The performance is further enhanced by an ensemble of both the three sub-models and the main model (92.79% and 71.86%). The improvement comes from the complementarity among the models, particularly between the models sharing a smaller number of layers. For instance, we found that sub-model 3 and the main model tend to show coincident classification results for a large portion of test data, so their complementarity is not significant; on the other hand, more data are classified differently by sub-model 1 and the main model, where we mainly observe performance improvement. Instead of the simple summation, there could be a better method to combine the models, which is left for future work.

## 4 CONCLUSION

In this paper, we proposed the local critic training approach for removing the inter-layer locking constraint in training of deep neural networks. In addition, we proposed three applications of the local critic training method: structural optimization of neural networks, progressive inference, and ensemble classification. It was demonstrated that the proposed method can successfully train CNNs with local critic networks having extremely simple structures. The performance of the method was also evaluated in various aspects, including effects of structures and update intervals of local critic networks and influences of the sizes of layer groups. Finally, it was shown that structural optimization, progressive inference, and ensemble classification can be performed directly using the models trained with the proposed approach without additional procedures.

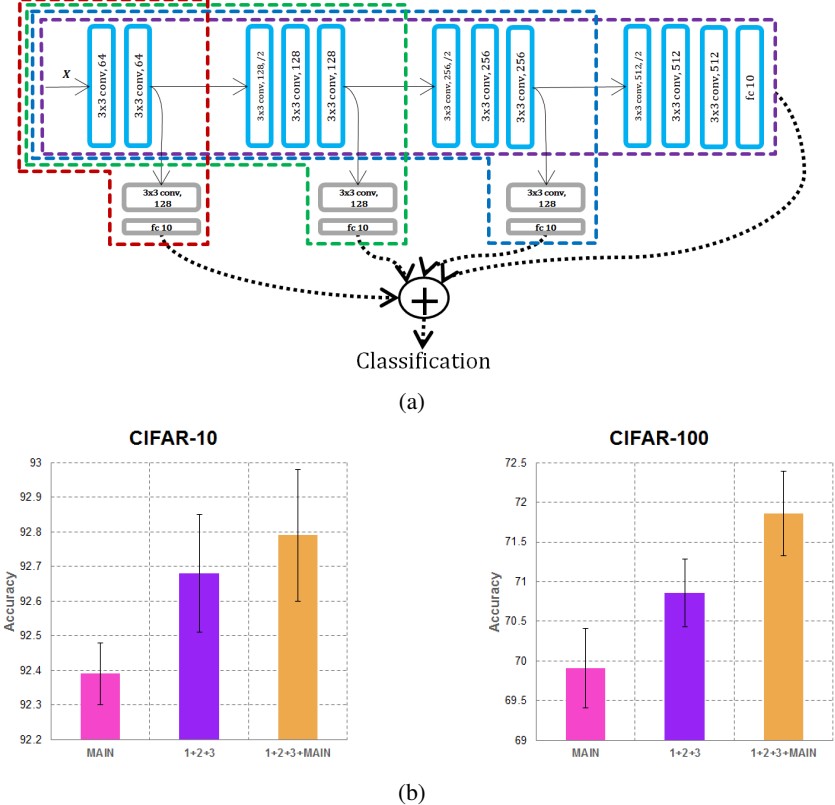

(a)

(b)

Figure 4: (a) Ensemble inference using the sub-models and main model. (b) Performance of the ensemble inference for an ensemble of the three sub-models (1+2+3) and an ensemble of the sub-models and the main model (1+2+3+main). Standard deviation values are also shown as error bars.

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

## A    ADDITIONAL RESULTS

### A.1    ADDITIONAL COMPARISON

In Czarnecki et al. (2017a), a method to minimize not only the loss of the network output but also its derivative is proposed, called Sobolev training, and applied to the critic training algorithm. We also conduct an experiment to use the Sobolev training method in our proposed algorithm. The results are shown in Table 7. In comparison to the performance shown in Table 1, we do not observe significant difference overall. In addition, we test the deep supervision algorithm (Wang et al., 2015), which also has additional modules connected to intermediate layers. The table shows that its performance is not significantly different from that of backpropagation.

Table 7: Average test accuracy (%) of Sobolev local critic training (Sob LC), Sobolev critic training (Czarnecki et al., 2017a), and deep supervision (Wang et al., 2015). The numbers of local networks used are shown in the parentheses. The standard deviation values are also shown.

| Dataset | Sob LC (1) | Sob LC (3) | Sob LC (5) | Sob critic (3) | Deep supervision (3) |
|---------|-----------|-----------|-----------|---------------|---------------------|
| CIFAR-10 | $91.83_{\pm 0.22}$ | $92.32_{\pm 0.12}$ | $91.42_{\pm 0.26}$ | $91.96_{\pm 0.22}$ | $94.08_{\pm 0.11}$ |
| CIFAR-100 | $73.64_{\pm 0.22}$ | $69.58_{\pm 0.32}$ | $63.82_{\pm 1.04}$ | $68.83_{\pm 0.31}$ | $75.09_{\pm 0.29}$ |

### A.2    RESULTS FOR LARGER NETWORKS

We examine the effectiveness of the proposed method for larger networks than those used in Section 3. For this, ResNet-50 and ResNet-101 (He et al., 2016a) are trained with backpropagation or the proposed method using three local critic networks. The results shown in Table 8 have a similar trend to those in Table 1 with slight performance improvement in most cases, which confirm that the proposed method works successfully for relatively complex networks.

Table 8: Test accuracy (%) of backpropagation (BP), and local critic training (LC) for ResNet-50 and ResNet-101. The numbers of local networks used are shown in the parentheses.

| Dataset | ResNet-50 BP | ResNet-50 LC (3) | ResNet-101 BP | ResNet-101 LC (3) |
|---------|-------------|-----------------|---------------|-------------------|
| CIFAR-10 | 94.42 | 92.95 | 94.18 | 93.17 |
| CIFAR-100 | 74.88 | 70.58 | 77.29 | 72.72 |

In addition, we experiment using the ImageNet dataset (Deng et al., 2009), which is much larger and more complex than CIFAR-10 and CIFAR-100. The results for ResNet-50 in Table 9 show that the proposed method can also work well for large datasets.

Table 9: Test accuracy (%) of ResNet-50 trained with backpropagation (BP) and local critic training (LC) for the ImageNet dataset.

| ImageNet | ResNet-50 BP | ResNet-50 LC (3) |
|----------|-------------|-----------------|
| Top-5 accuracy | 92.38 | 86.52 |
| Top-1 accuracy | 75.07 | 65.41 |

## B    VISUALIZATION OF TRAINED MODELS

In order to analyze the trained networks, we obtain the representational dissimilarity matrix (RDM) (Kriegeskorte et al., 2008) from each layer of the networks trained by backpropagation and the proposed method. For each of 400 samples from CIFAR-10, the activation of each layer is recorded, and the correlation between two samples is measured, which is shown in Figure 5. In the figure, clear diagonal-blocks indicate that samples from the same class have highly correlated representations (e.g., the last layer).

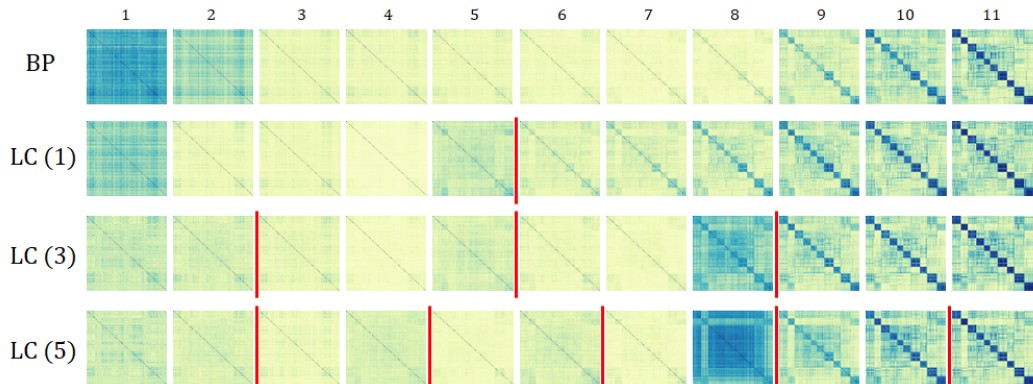

Figure 5: Representation dissimilarity matrix of each layer of the trained networks (having the structure shown in Figure 2a) for label-ordered samples from CIFAR-10. Red lines indicate the locations of local critic networks. A clear diagonal-block pattern indicates that clear inner-class representation has been trained.

Overall, block-diagonal patterns become clear at the last layers for all cases. However, the figure also shows that the two training methods result in networks showing qualitatively different characteristics in their internal representations. In particular, the layers at which local critic networks are attached (e.g., layer 5 in LC (1) and LC (3), and layer 6 in LC (5)) show relatively distinguishable block-diagonal patterns in comparison to those of the network trained by backpropagation. These layers in the proposed method act not only as intermediate layers of the main network but also as near-final layers of the sub-models, and thus are forced to learn class-specific representations to some extent.

## C  LEARNING DYNAMICS

### C.1  VISUALIZATION OF LOSS EVOLUTION

As a way to investigate the learning dynamics of the proposed method, the loss values at each local critic network ($L_i$) for individual data over the training iterations are examined (Czarnecki et al., 2017b). Figure 3 visualizes the loss values for 400 sampled data of CIFAR-10 (arranged in 2D) when three local critic networks are used. For visualization, the same results are shown twice, once sorted by labels (Figure 6a) and once sorted by $L_4$ at iteration 20000 (Figure 6b). At the early stage of learning, the loss values at the local critic networks ($L_1$ to $L_3$) are largely different from those at the main network ($L_4$), with only slight similarity (e.g., the blue region at iteration 50 in Figure 6a). At the later stage, however, all the losses similarly converged to small values for most of the samples (at iteration 20000 in Figure 6b).

### C.2  LEARNING CURVES OF SUB-MODELS

We showed the test performance of sub-models in Table 4. In addition, we show their test accuracy over training iterations for CIFAR-10 in Figure 7. In particular, faster convergence in the case of local critic training than backpropagation is observed in Figure 7a.

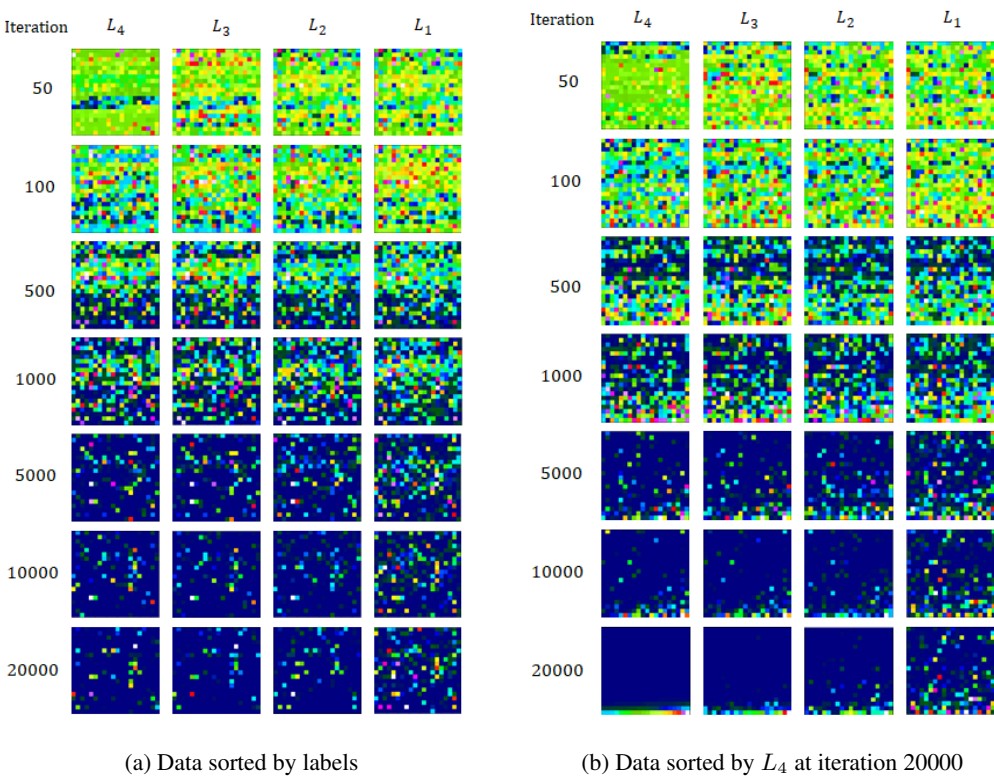

(a) Data sorted by labels                    (b) Data sorted by $L_4$ at iteration 20000

Figure 6: Evolution of the loss value during the course of the proposed local critic training.

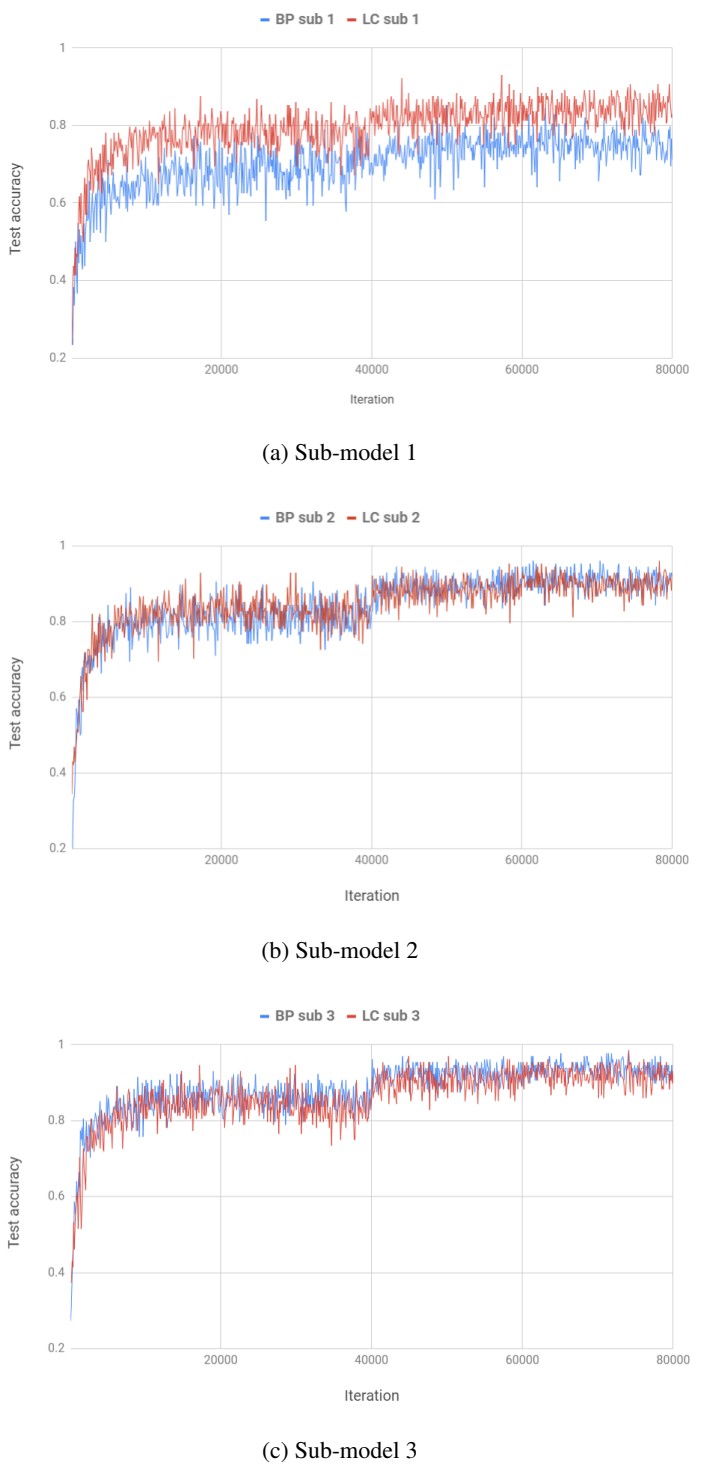

(a) Sub-model 1

(b) Sub-model 2

(c) Sub-model 3

Figure 7: Test accuracy (%) of sub-models trained by backpropagation and local critic training over training iterations.

