# OpenReview forum: "Local Critic Training of Deep Neural Networks"
_ICLR.cc/2019/Conference_

### Official Review · AnonReviewer3 · 2018-11-01
**Local extension of critic training**

**Rating:** 7
**Confidence:** 5

**Review:**

This paper describes a method of training neural networks without update locking. The idea is a small modification on top of Czarnecki et al. Critic training, where instead of using final loss as a critic target, one bootstraps from critics on other layers. In particular, if only one module is present, these two approaches are actually identical. To be more precise, the only difference between these two methods is that (7) in Critic training would change to l(L_i, L_N). As a consequence, method becomes forward unlocked too. It is worth noting, that in the appendix of Czarnecki et al. it is shown that this particular method (critic training) under simple conditions actually "degenerates" to deep supervision (which is forward unlocked too). Consequently unlocking property as such is not a big contribution of the proposed method. Rest of the paper includes following elements:
- empirical evaluation showing improvement over critic training by 0.4% in CIFAR10 and 0.9% in CIFAR100 when using 3 splits.
- expansion on using the model for progressive inference.

Given standard deviation of errors in Table 1. it is not clear how significant these improvements are. How many samples were used to estimate these quantities? It is worth noting, that Critic training was showed to be outperformed by Sobolev Training in the same paper authors cite, but its performance is not reported despite looking like a well defined baseline. In particular, can these two methods be combined?

I believe that this is an interesting research direction, however paper in its current form seems as a small incremental improvement over sota, and could be significantly improved by for example:
- providing more comprehensive evaluation (including estimating accuracy to lower std errors)
- adding other baseline solutions (such as Sobolev training, cDNI, or deep supervision)
- considering any form of convergence/dynamics analysis of the proposed approach

---

> ### Author Response · Authors · 2018-11-18
> **Authors' reply**
>
> We thank the reviewer for the valuable feedback!
>
> We conducted several additional experiments and added the results as the supplementary material. Please check the revised paper.
>
> 1. Unlocking property
> In the deep supervision, a certain layer is trained in the direction to minimize the weighted sum of the main loss (loss of the main network) and the supervision loss (loss of the additional modules located above the layer), therefore training of the layer is possible only after the feedforward pass finishes and then the error gradient is backpropagated.
> In the critic training, training of the additional modules requires L_N, and thus needs to wait until the feedforward pass completely ends to obtain the final output of the main network; in this sense, the method is not fully unlocked.
> On the other hand, in our approach, training of a layer does not need to wait until the main network's output is obtained and then the error gradient is backpropagated till the layer, and furthermore, training of a local critic network does not need to wait until the main network's output is obtained. Therefore, our method achieves the most complete unlocking property. Thus, we believe that our modification brings a significant opportunity for training with unlocking.
>
> 2. Comprehensive evaluation (improvement over critic training)
> All the experiments are performed 5 times (mentioned in the sentence right before Section 3.1), from which the average and standard deviation values are obtained.
> We conducted nonparametric Mann-Whitney tests in order to examine statistical significance of the improvement of our method over critic training (0.4% and 0.9% for CIFAR-10 and CIFAR-100, respectively), from which we found the improvement is significant. This result is added in the revised paper, Section 3.1.
>
> 3. Other baselines
> We ran more experiments for other baselines (Sobolev training applied to both our method and the critic training method, and deep supervision). The results are provided in Section A.1 of the supplementary material.
>
> 4. Convergence/dynamics analysis
> We added analysis of learning dynamics of the proposed approach in Section C of the supplementary material.
>
> We hope that the revised paper clears all the concerns the reviewer raised.

---

> > ### Comment · AnonReviewer3 · 2018-11-18
> > **Thanks you for the revision**
> >
> > Thank you for providing revised version of the paper. I am increasing my initial evaluation to 7: Good paper, accept, given the updates, especially wrt. evaluation protocol.

---

### Official Review · AnonReviewer1 · 2018-11-06
**review of local critic training of deep neural networks**

**Rating:** 6
**Confidence:** 4

**Review:**

This manuscript presents a new method that can conduct local training of a deep neural network.
Briefly speaking, the proposed method first cuts a very deep network into a few groups and then train the parameters of each group almost independent of the other groups. The main idea is to attach a local critic module to each group and the error gradient is back propagated to each group from its local critic module instead of the last layer of the whole network.
Such an idea seems to work well and the resultant performance decrease is acceptable when the original network is not very large. However, when the original network is complex, the performance decrease may be relatively big.

Another benefit of this local critic training is that in addition to the main model, it can also produce several submodels that can be used for ensemble inference and progressive classification.

In summary, this manuscript proposes an interesting idea, but not sure empirically how useful it will be since for a complex network, this method may result in relatively big performance decrease. For a simple network, although the performance decrease is small, but there is no need to use the proposed training method for a simplex network.

---

> ### Author Response · Authors · 2018-11-18
> **Authors' reply**
>
> We thank the reviewer for the valuable feedback!
>
> We conducted several additional experiments and added the results as the supplementary material. Please check the revised paper.
>
> In particular, the reviewer questioned about the performance of the proposed method for more complex networks. To examine that, we additionally conducted experiments with larger networks (ResNet-50 and ResNet-101). The results are reported in Section A.2 of the Supplementary Material. When the results in Table 1 and Table 8 are compared, it can be observed that the proposed LC method works well even for large networks without performance degradation.
>
> We hope that the revised paper clears all the concerns the reviewer raised.

---

> > ### Comment · AnonReviewer1 · 2018-11-18
> > **on more complex networks**
> >
> > I appreciate that the authors provided more experimental results, which further confirmed that the performance decrease is big when the proposed idea is applied to a large network.  That is, it is hard to tell if the proposed method will be empirically useful or not.

---

> > > ### Author Response · Authors · 2018-11-27
> > > **Authors' reply**
> > >
> > > Thank you again for the feedback!
> > >
> > > We improved the performance for large networks and updated the results in Tables 8 and 9 in Section A.2 of the Supplementary Material. For ResNet-101, we adjusted the locations of the local critic networks so that they are placed evenly along the main network (previously, the local critic networks were placed where downscaling occurs, so the number of layers for which each local critic network serves was largely different); for ImageNet, we optimized the learning parameters (in particular, the weight decay parameter). Now, the difference of the top-5 accuracy between backpropagation and our method for ImageNet is only 5.86%, which is similar to that for CIFAR-100 in the main paper. And, the difference of the accuracy between backpropagation and our method for CIFAR-100 using ResNet-101 is only 4.57%, which is even smaller than that for CIFAR-100 using the small network in the main paper.
> > >
> > > Thus, we do not observe particularly big performance decrease for large networks.

---

### Official Review · AnonReviewer2 · 2018-11-12

**Rating:** 6
**Confidence:** 3

**Review:**

This paper proposes an alternative training paradigm for DNIs: instead of training an auxiliary module to approximate the gradient provided by the following modules of the original model, they train it to approximate directly the final output of the original model. Although the approach does not seem to improve significantly, if at all, over Sobolev training of these modules (denoted 'critic' in the paper), this method seems simpler and offer side benefits which seem to be the main contribution of this paper.
Indeed Table 4 and 5 show for example how they can, after training the full model, extract a submodel requiring significantly less computation and parameters with little loss in the performance. Alternatively, they also propose a method to do progressive inference for similar reasons.
The ensembling result is interesting, but the figure 4 is not necessarily clear on the significance of this result, especially since the standard deviation is not shown in this figure.
The idea proposed in this paper is interesting, however, the experiments are restricted on relatively small and simple architectures and limited on two very similar datasets (CIFAR-10 and CIFAR-100), making the argument less compelling.

---

> ### Author Response · Authors · 2018-11-18
> **Authors' reply**
>
> We thank the reviewer for the valuable feedback!
>
> We conducted several additional experiments and added the results as the supplementary material. Please check the revised paper.
>
> 1. Figure 4
> We revised Figure 4 to include standard deviations. It appears that the improvement due to ensemble inference is significant in most cases.
>
> 2. Architecture and dataset
> We additionally conducted experiments with larger networks (ResNet-50 and ResNet-101) and with a larger dataset (ImageNet). The results are reported in Section A.2 of the Supplementary Material. In short, the results show that the proposed method still works for these large networks and large dataset.
>
> We hope that the revised paper clears all the concerns the reviewer raised.

---

### Meta-Review · Area_Chair1 · 2018-12-13

**Confidence:** 2
**Recommendation:** Reject

**Metareview:**

This paper proposes a new training approach for deep neural interfaces. The idea is to bootstrap from critics of other layers instead of using the final loss as target. The method is evaluated of CIFAR-10 and CIFAR-100 and found to improve performance slightly upon Sobolev training while being simpler. The reviewers found the idea interesting but were concerned about the strength of the experimental results. The datasets are similar and the significance of the results is not clear. The revision submitted by the authors was only able to address some of these issues such as the evaluation protocol.